# Fecal Bile Acids and Neutral Sterols Are Associated with Latent Microbial Subgroups in the Human Gut

**DOI:** 10.3390/metabo12090846

**Published:** 2022-09-08

**Authors:** Taylor A. Breuninger, Nina Wawro, Dennis Freuer, Sandra Reitmeier, Anna Artati, Harald Grallert, Jerzy Adamski, Christa Meisinger, Annette Peters, Dirk Haller, Jakob Linseisen

**Affiliations:** 1Chair of Epidemiology, University Hospital Augsburg, University of Augsburg, Stenglinstr. 2, 86156 Augsburg, Germany; 2Helmholtz Zentrum München, German Research Center for Environmental Health (GmbH), Institute of Epidemiology, Ingolstädter Landstr. 1, 85764 Neuherberg, Germany; 3Chair of Nutrition and Immunology, Technische Universität München, Gregor-Mendel-Str. 2, 85354 Freising, Germany; 4ZIEL—Institute for Food & Health, Technische Universität München, Weihenstephaner Berg 3, 85354 Freising, Germany; 5Helmholtz Zentrum München, German Research Center for Environmental Health (GmbH), Metabolomics and Proteomics Core, Ingolstädter Landstr. 1, 85764 Neuherberg, Germany; 6Institute of Experimental Genetics, Helmholtz Zentrum München, German Research Center for Environmental Health, Ingolstädter Landstraße 1, 85764 Neuherberg, Germany; 7Department of Biochemistry, Yong Loo Lin School of Medicine, National University of Singapore, 8 Medical Drive, Singapore 117597, Singapore; 8Institute of Biochemistry, Faculty of Medicine, University of Ljubljana, Vrazov trg 2, 1000 Ljubljana, Slovenia

**Keywords:** fecal metabolites, bile acids, gut microbiota, 16S rRNA gene sequencing

## Abstract

Bile acids, neutral sterols, and the gut microbiome are intricately intertwined and each affects human health and metabolism. However, much is still unknown about this relationship. This analysis included 1280 participants of the KORA FF4 study. Fecal metabolites (primary and secondary bile acids, plant and animal sterols) were analyzed using a metabolomics approach. Dirichlet regression models were used to evaluate associations between the metabolites and twenty microbial subgroups that were previously identified using latent Dirichlet allocation. Significant associations were identified between 12 of 17 primary and secondary bile acids and several of the microbial subgroups. Three subgroups showed largely positive significant associations with bile acids, and six subgroups showed mostly inverse associations with fecal bile acids. We identified a trend where microbial subgroups that were previously associated with “healthy” factors were here inversely associated with fecal bile acid levels. Conversely, subgroups that were previously associated with “unhealthy” factors were positively associated with fecal bile acid levels. These results indicate that further research is necessary regarding bile acids and microbiota composition, particularly in relation to metabolic health.

## 1. Introduction

The complex interplay between bile acids and the human gut microbiome is emerging as an important factor in the regulation of many physiological and metabolic processes. Traditionally, the main importance of bile acids in human physiology was ascribed to their role in the digestion and absorption of fats. The primary bile acids in humans are cholic acid and chenodeoxycholic acid, usually conjugated with glycine, taurine, or sulfate [1,2]. After secretion into the small intestine, bile acids act as emulsifiers, and approximately 95% of bile acids are reabsorbed from the small intestine and returned to the liver via the portal vein, where their presence regulates the de novo production of cholesterol and primary bile acids. However, efficient as this process is, a small amount of bile acids is not reabsorbed and passes into the large intestine, where they are transformed by the gut microbiota into secondary bile acids [3,4]. Secondary bile acids have even stronger antimicrobial properties than primary acids. Thus, through both direct and indirect cytotoxic and antimicrobial effects, bile acids in turn regulate the composition of the gut microbiome itself [5,6].

While the role of the gut microbiota in transforming primary to secondary bile acids has been known for some time, it has only become clear in recent decades how large of an impact the gut microbiota has on the health of its human host. Some of the effects may be mediated by (secondary) bile acids [3]. The gut microbiota is now sometimes considered a metabolic organ in its own right due to its important role in modulating host health and metabolism, largely mediated through its production of a variety of metabolically active compounds [3,4]. These compounds include bile acids, which regulate metabolic homeostasis, glucose, and, in particular, lipid metabolism through the activation of farnesoid X receptor (FXR, which is also found in the intestine) and G protein-coupled receptors [7,8]. Neutral sterols, including those of both plant and animal origin, are compounds which are ingested through diet and which may also be modified by gut microbiota or play a role in shaping gut microbiota composition [9]. In recent years, the composition of the gut microbiota has been associated with a plethora of disease states including obesity, cardiovascular disease, irritable bowel disease, and many others [10,11,12,13]. However, it has thus far proven difficult to translate this knowledge into concrete treatments or recommendations, as much remains unknown despite nearly two decades of intense research interest.

Plant and animal sterols also interact with the gut microbiome and may even play an important role in shaping gut microbiota composition. While cholesterol metabolism in the microbiota has been studied relatively well, the interaction between plant sterols and the microbiota has not received such attention [14]. However, it is not entirely clear which bacteria are involved in the intestinal transformation of cholesterol [15]. Furthermore, very little is known about the metabolism of plant sterols by the microbiota or how they may affect microbiota composition.

In a previous study, we sought to examine the complex relationship between diet, the gut microbiome, and metabolic diseases or risk factors in a population-based cohort [16]. We used the machine learning method latent Dirichlet allocation to identify twenty latent microbial subgroups within the study population, a unique combination of which can be used to describe each individual’s microbiota composition. We then identified associations between habitual dietary intake and metabolic diseases or risk factors and the subgroups. Many of the subgroups were significantly associated with habitual diet and/or metabolic diseases or risk factors, not only confirming suitability of the method for use with microbiome data, but also indicating biological plausibility and potential relevance of these subgroups in disease etiology. In the present study, we seek to investigate the relationship between fecal bile acids and sterols and these twenty subgroups. 

## 2. Materials and Methods

### 2.1. Study Population

This cross-sectional analysis was conducted in the Cooperative Health Research in the Augsburg Region (KORA) FF4 (2013/2014) study population. The KORA FF4 study is the second follow-up of the S4 survey (1999/2001), one of four population-based studies (MONICA S1–S3, KORA S4) which recruited participants from Augsburg and its two surrounding counties with a focus on studying diabetes and cardiovascular risk factors. In the KORA FF4 study, 1280 participants (of 2279 total) had both gut microbiome and metabolite data available and did not take antibiotics in the two months leading up to sample collection. Sub-analyses including diet variables were restricted to 930 participants due to missing dietary intake data. Further details regarding the KORA studies have been described previously [17]. This study was conducted in accordance with the Declaration of Helsinki. All procedures involving human subjects were approved by the ethics committee of the Bavarian Chamber of Physicians in Munich. All participants gave their written, informed consent.

### 2.2. Collection of Biosamples and Participant Data

On the day of each individual’s study center visit, anthropometric measurements were taken in a standardized fashion by a trained examiner. A detailed questionnaire regarding participants’ lifestyle, disease history, and medication use was administered by a trained examiner. Data on habitual dietary intake were gathered via one food frequency questionnaire and up to three repeated 24 h food lists. Participants collected stool samples at home, on the day of the study center visit when possible, using a kit they received by mail. Each kit included two collection tubes, one containing 5 mL of DNA stabilizer (Stratec DNA Stool Stabilizer, No. 1038111100) and the other without. One spoonful sampled from two areas of the stool specimen were deposited into each tube. Participants completed a questionnaire they received with the kit and kept the sample refrigerated until transportation to the study center. Samples were excluded if patients did not follow instructions properly, if the sample was left unrefrigerated for more than 3 h, or if the patient reported taking antibiotics within the previous 2 months. After arrival at the study center, samples were stored initially at −20 °C and eventually −80 °C. 

### 2.3. Metabolomics Analysis

The stool sample that was not preserved in DNA stabilizer was used for the metabolomics analysis. The details of the metabolomics analysis have been described previously [18]. Briefly, the frozen samples were diluted with pre-cooled water in a pre-cooled homogenization tube and homogenized in a machine equipped with a cooling unit. Dry mass was then determined and a 100 µL aliquot of the homogenate was pipetted into a 2 mL 96-deep well plate. Six additional wells were filled with human stool reference, one with human plasma reference, and a further six with water as process blanks (references from Seralab, West Sussex, UK). The samples were then extracted with methanol containing four recovery standards, centrifuged, split into four aliquots each, dried, and reconstituted with a solvent compatible with all four methods. Two of the aliquots were analyzed by reverse phase ultra-high-performance liquid chromatography/tandem accurate mass spectrometry (RP/UPLC-MS/MS) with positive ion mode electrospray ionization (ESI). One was optimized for hydrophilic compounds and the other for hydrophobic compounds. The third aliquot was analyzed by RP/UPLC-MS/MS with negative ion mode ESI, and the fourth was analyzed by hydrophilic interaction liquid chromatography (HILIC)/UPLC-MS/MS with negative ion mode ESI. The MS analysis alternated between MS and data-dependent MS2 scans using dynamic exclusion, and the scan range was 70–1000 m/z. The metabolites were extracted and profiled by Metabolon, Inc. (Durham, NC, USA). The metabolites were identified by automated comparison of the ion features in the experimental samples to a reference library of chemical standard entries that included retention time, molecular weight (m/z), preferred adducts, and in-source fragments as well as associated MS spectra, and were then curated by visual inspection for quality control using software developed at Metabolon.

### 2.4. 16S rRNA Gene Amplicon Sequencing and Amplicon Analysis

The stool samples preserved in DNA stabilizer were used for high-throughput 16S rRNA gene amplicon sequencing. The methods have already been described in detail by Reitmeier et al. [19]; however, briefly, the cells were lysed using a bead-beater with 0.1 mm glass beads, and DNA was purified and extracted. DNA was diluted in water and used as a template for 25 cycles of amplification of the V3–V4 regions of 16S rRNA genes. PCR-fragment concentration was determined by fluorometry and adjusted to 2 nM. PCR products were then pooled during cleaning with magnetic beads. Two negative controls and one positive control were used in each batch of 45 samples (processed on one 96-well plate). The multiplexed samples were sequenced on an Illumina HiSeq in paired-end mode (2 × 250 bp) using Rapid v2 chemistry. Illumina MISeq using v3 was used to re-sequence any samples with read counts below 4700. Analyses were based on chimera-checked, high-quality sequences. 

The IMNGS pipeline was used for preprocessing of the sequencing data [20]. Chimera were removed using UCHIME [21]. UPARSE v8.1.1861_i86 was used to merge de-multiplexed reads and cluster by 97% similarity. A relative abundance of <0.25% across samples was set as the cut-off level to exclude spurious OTUs from the analysis [22]. The RDP classifier version 2.11 and the SILVA database were used to assign taxonomy [23]. 

### 2.5. Calculation of Habitual Dietary Intake

The calculation of habitual dietary intake in the KORA FF4 study has been described previously [24]. Briefly, dietary intake was assessed via the means of one food frequency questionnaire (FFQ) and two to three repeated 24 h food lists (24HFLs). A two-step method incorporating both instruments was used to calculate habitual dietary intake. In the first step, consumption probability on a given day was determined for each food based on the 24HFLs and FFQ. In a second step, the usual portion size for each item for each person was determined based on data from the Bavarian Food Consumption Survey II (BVSII). Final habitual dietary intake for a given food on a given day was then calculated by multiplying consumption amount by consumption probability. The European Prospective Investigations into Cancer and Nutrition (EPIC) Soft classification scheme was used to categorize food items into 16 food groups [25], and the National Nutrient Database (Bundeslebensmittelschlüssel; BLS 3.02) was used to calculate habitual nutrient intake.

### 2.6. Identification of Microbial Subgroups/Latent Dirichlet Allocation

The identification of the 20 microbial subgroups was described in detail previously [16]. In summary, the machine learning method latent Dirichlet allocation (LDA) was applied to the normalized OTU table in order to detect hidden structures within the data frame. LDA is a popular Bayesian probabilistic generative machine learning model that can identify hidden structures of co-occurring features in a dataset. It is most commonly used in the field of natural language processing, for example, to identify article topics among a dataset of text documents such as “politics”, “religion”, and “sports” [26]. However, it has been successfully implemented in a number of biological areas as well [27,28]. When applied to microbiome data, LDA is able to detect latent microbial subgroups of bacteria that tend to co-occur across samples. Although LDA determines patterns of co-occurrence, rather than using a distance measure as in clustering, the subgroups identified by LDA can be compared to clusters with fractional membership. Each individual’s microbiota composition can be described with a unique combination of all subgroups. For example, one individual’s microbiome may be described 70% by one subgroup, 20% by another, 5% by a third subgroup, and 5% by all remaining subgroups combined, whereas another individual’s microbiota may be described approximately 10% by 10 different subgroups. Each percentage is actually a probability indicating how likely it is that an individual’s microbiota contains the respective subgroup. The subgroups are specific to the population in which they were identified, and one unique combination of all 20 subgroups is identified to describe each individual’s microbiome.

Likewise, each subgroup can be described by a unique combination of all OTUs. In the same manner as described above, different subgroups may be described by higher or lower probabilities of containing each OTU in the dataset. Some subgroups may be heavily dominated by one or several OTUs, whereas others may be described with equal probabilities of many OTUs. In learning patterns of co-occurrence, each subgroup then represents a unique group of microbes that tend to appear together for one reason or another, such as similar environmental requirements, similar functions, or some other unknown factor. 

After applying exclusion criteria (no antibiotic intake in previous 2 months), 1992 participants had microbiome data available. The OTU table was filtered to a relative abundance of >0.1% and a prevalence of 1%, which resulted in 1713 OTUs which were used in the analysis. The number of subgroups (*n* = 20) was chosen due to practicality as perplexity and log likelihood continually improved with as the number of subgroups increased. The LDA model was implemented using the R package MetaTopics version 1.0 [29].

### 2.7. Metabolite–Subgroup Associations

The associations between fecal sterols or bile acids and the microbial subgroups was assessed via Dirichlet regression models using the common parametrization (R package DirichletReg version 0.7-1). Dirichlet regression is a type of regression model allowing the outcome variable to be a matrix of multiple compositional variables (as is the case for the microbial subgroups) [30]. Any missing values in the fecal metabolite variables were imputed with the minimum value within each metabolite. Any variables with over 25% missing values were excluded from the analysis. A directed acyclic graph was used to identify age, sex, BMI or waist circumference, physical activity, fiber, fat, energy intake, and alcohol as potential confounding variables to be used for adjustment. As fat and energy intake were highly correlated, fat was not included in the models. The dietary variables (fiber, fat, energy, and alcohol) were only included in a sub-analysis due to the limited sample size (missing dietary data). All continuous variables were standardized before being entered into the model. The model for each category of metabolite was tested for improved fit with a quadratic term using ANOVA. If the models were significantly improved, the quadratic term was left in the models. A sub-analysis in participants with information on habitual dietary intake was performed in the same manner, but models were additionally adjusted for total energy intake, total fiber, and alcohol intake. An additional sensitivity analysis was also performed, excluding all participants taking statins or other lipid-lowering medications. All models were adjusted for multiple testing using the Bonferroni correction (0.05/24 = *p* < 0.0021).

### 2.8. Descriptive Statistics and Figures

All statistical analyses were done in the software R (Version 4.1.2) using R studio (2021.09.2 Build 382). For Table 1, mean and standard deviation were calculated for continuous variables, and proportion and frequency were calculated for categorical variables. The minimum, 25th percentile, median, 75th percentile, maximum, and percentage missing vales were calculated as shown in Table 2. The R package ggplot2 (version 3.3.5) was used to create Figure 1 and Figure 2; the package corrplot (version 0.92) was also used to create Figure 2 [31].

## 3. Results

Table 1 describes the study population as a total and according to sex by age (years), waist circumference (cm), education level (<13 years, ≥13 years), and physical activity levels (active > 1 h per day, inactive) as well as habitual daily intake of total energy (kcal), total fiber (g/d), and alcohol (g/d). Participants range in age from 38 to 87, with a mean age of 59.72 years; 50.47% identify as male and 49.5% identify as female. The average waist circumference for men is 102 cm (±11.5), with 91.51 cm (±13.9) for women. The majority of men and women had <13 years of education, although more men than women (41.8% vs. 30.3%) had ≥13 years of education. A larger percentage of women than men were physically active for at least one hour daily (60.3% vs. 55.7%). Men had a higher calorie intake and slightly higher mean fiber intake and also consumed more alcohol than women.

Descriptive statistics for each metabolite before imputation or transformation are given in Table 2. The metabolite intensities minimums range from 0.0002 to 0.003 and the maximums range from 0.465 to 64.81. The median intensities range between 0.048 and 0.055. The percentage of missing values ranges from 0.23% for cholesterol to 21.25% for 7,12-diketolithocholate. 

The composition of the 20 microbial subgroups is depicted in Figure 1. The 39 genera assigned a probability of at least 0.02 for at least one subgroup are shown in the figure. All genera below this threshold were collapsed into an “Other” genus, which contributes 1.37–8.61% of each subgroup. All OTUs that could not be identified at the genus level were collapsed into an “NA” genus. Here, it is easy to see that certain subgroups are dominated by one genus (e.g., subgroups 3, 4, 9, 13, 15, 19), while others are a more homogenous mix of several genera (e.g., subgroups 1, 10, 11, 12, 14). Several subgroups contain mostly unidentified taxa (e.g., subgroups 14 and 20). 

The results of the Dirichlet regressions between neutral sterols or bile acids and the microbial subgroups are displayed graphically in Figure 2. Subgroups 1, 4, 5, 6, 10, 12, 13, 14, 16, 18, and 20 were significantly associated with one or more metabolite after adjustment for multiple testing. Cholesterol was significantly (positively) associated with subgroups 6 and 13, and inversely associated with subgroups 1, 5, 10, 14, 16, 17, 18, and 20. Coprostanol was significantly associated with the same groups as fecal cholesterol (with the exception of subgroups 5 and 10), but in the opposite direction. 

The plant sterol models showed indication of a non-linear relationship, and the results of the models including a quadratic term can be found in Appendix A. However, as it is not possible to display both a linear and quadratic beta and *p*-value for these models in the heat map, the results for the plant models show the linear trend for these associations. The non-linear associations in comparison with the linear trends for these models can be viewed in Appendix A. Significant associations were identified between plant sterols and Subgroups 1, 6, 10, 13, 14, 16, 17, 18, and 20. Associations between stigmasterol, campesterol, and beta-sitosterol and the subgroups were consistent in direction, while associations between sitostanol and the subgroups were always in the opposite direction. The strongest associations among plant and animal sterols were with subgroup 20. Only ergosterol was not associated with any subgroup according to the linear trend results (shown in heat map). However, the non-linear models (Appendix A) show a differential association between ergosterol and the subgroups, where the linear beta is negative and the quadratic beta is positive, and the majority of the associations are statistically significant. 

Subgroups 1, 5, 6, 14, 16, and 20 had the strongest and most numerous associations with metabolites, particularly with secondary bile acids. Subgroups 1, 5, 14, 16, 18, and 20 were largely negatively associated with primary and/or secondary bile acids in stool, with the exception of dehydrolithocholate, which was significantly positively associated with subgroups 1, 18, and 20. Subgroups 4, 6, 12, 13, and 18 were significantly positively associated with one or more primary or secondary bile acid. Appendix A contains the results displayed in Figure 2 in table form.

The results of a sub-analysis of the Dirichlet regression models additionally adjusted for total energy, fiber, and alcohol intake are listed in Appendix A. Although a few of the associations lost significance, more frequently with the plant sterols and subgroups 9, 10 or 17, the majority did not change or the significance of the association became more pronounced. 

A sensitivity analysis of the Dirichlet regression models was also performed, in which participants taking lipid-lowering medications or statins (*n* = 208) were excluded from the analysis in order to assess potential effects of these medications on bile acid or sterol levels. The results are shown in Appendix A. For the most part, results remained the same. However, a handful of associations lost their significance, and a few newly became significant. Overall, the direction and strength of the associations remained consistent.

## 4. Discussion

In the present study, we used fecal metabolomics data to analyze the association between bile acids and other sterols and microbiota composition. The latter was described by 20 microbial subgroups, each composed of a unique combination of taxa at the genus level [16]. We observed that the fecal bile acids are differentially associated with these microbiota subgroups. The subgroups which were associated with a healthier dietary intake (and/or fewer metabolic diseases) were inversely associated with fecal bile acids, while those subgroups which were previously associated with an unhealthier dietary behavior (or metabolic diseases/risk factors) were typically positively associated with fecal bile acids. With these differential associations, we support the biological validity of the microbiota subgrouping concept. 

Indeed, subgroups 5, 14, and 16 were the subgroups which, in our previous analysis, were most strongly (positively) associated with health-promoting dietary factors and inversely associated with metabolic risk factors [16]. In the present analysis, subgroups 5, 14, and 16 were also the subgroups most strongly and consistently significantly inversely associated with primary and secondary bile acids. Subgroups 1, 18, and 20 were also inversely associated with 6, 1, and 8 of the 14 secondary bile acids, respectively. However, interestingly, they were also significantly positively associated with dehydrolithocholate, indicating a possible differential relationship between dehydrolithocholate and other bile acids with these subgroups. Subgroups 1 and 20 were also inversely associated with cholate. In our previous study, subgroups 1, 18, and 20 were also inversely associated with metabolic diseases or risk factors, though regarding diet, they were only (positively) associated with wine, sweets, and soluble fiber, respectively [16]. These favorable subgroups seem to flourish in the presence of lower levels of bile acids in stool. These subgroups may be particularly sensitive to the antimicrobial effects of bile acids, or a more indirect effect may be present, such as the potential modulation of both these subgroups and bile acids through another unidentified factor. Further studies should seek to identify potential mechanisms.

Subgroups 6, 12, and 13, on the other hand, were significantly positively associated with 8, 2, and 5 primary and/or secondary bile acids, respectively. It appears that *Bacteroides* species may play an important role in the deconjugation of primary bile acids [32,33]. Indeed, subgroups 6 and 13, among a few others, are dominated by the genus *Bacteroides* (52.31% and 74.76%, respectively) [16]. *Bacteroides* and in particular *Bifidobacterium*, the latter of which makes up a moderate amount of subgroup 12 (13.40%), are also among the bacteria which are known to produce bile salt hydrolase, responsible for the deconjugation of secondary bile acids which may modulate gut microbiota composition through increased antimicrobial activity and increase bile acid excretion in stool [32,34]. These mechanisms may contribute to the positive associations between primary and/or secondary bile acids and these subgroups in our study. In our previous study, subgroups 6 and 13 were positively associated with serum triglycerides and BMI, respectively, while subgroup 12 was inversely associated with the Alternate Healthy Eating Index 2010 (AHEI), a dietary index where a higher score correlates to higher diet quality. Here, these three subgroups which were previously associated with unfavorable health aspects (be it diet quality or metabolic risk factors) are positively associated with fecal bile acids. Bile acids are important regulators of metabolism through a variety of mechanisms, particularly through their action as agonists of farnesoid X receptor (FXR) and the G-protein coupled receptors Takeda G-protein coupled receptor (TGR5) and sphingosine-1-phosphate receptor 2 (S1PR2). These receptors play an important role in regulating energy, glucose, and lipid metabolism [7,8]. It is therefore to be expected that bile acid levels in feces are associated with the same microbial subgroups that were previously associated with metabolic diseases or risk factors, and it is possible that interactions between the microbiota and bile acids through these mechanisms mediate the previously identified associations between subgroups 6 and 13 and metabolic outcomes [2]. Likewise, bile acid metabolism is closely linked with diet, as dietary intake determines bile acid production and release, while bile acids themselves affect the metabolism and absorption of (lipids and lipid-soluble) nutrients. Our results underline that gut microbiota composition, fecal bile acid levels, dietary intake, and metabolic health are closely intertwined. 

Bile acids have been associated with both positive and negative effects on the gut microbiota through numerous mechanisms. For example, a decrease in the secondary bile acid pool has been implicated in increased susceptibility to colonization by pathogenic bacteria, likely due to the loss of the antimicrobial effects of secondary bile acids in the gut [35]. Reduced levels of bile acids in the gut have also been associated with inflammation and dysbiosis [36]. However, a number of pathogenic bacteria, including *Salmonella* spp and *Escherichia coli* employ several strategies to make them resistant to bile, and, in fact, their growth may be encouraged by high levels of (primary) bile acids due to their advantages over commensal bacteria in the microbiota. Indeed, bile acid exposure has been demonstrated to increase virulence of these pathogens as well as that of *Shigella* spp. In this way, bile acids (especially more toxic secondary bile acids) may be protective against bacterial pathogens and potentially even viruses in the gut [1]. Additionally, irritable bowel diseases are typically associated with dysbiosis and impaired bile acid absorption, resulting in increased levels of bile acids in the lumen [1]. However, studies have demonstrated that higher levels of secondary bile acids are associated with remission of ulcerative colitis after fecal microbial transplantation, likely due to a more diverse and robust microbiome (and therefore increased conversion to secondary bile acids) [37].

Yet high levels of secondary bile acids in the gut have also been linked with dysbiosis and intestinal and hepatic carcinogenesis [1]. The regulation of bile acid metabolism and the bile acid pool is an intricate and delicate process, affected by many factors including the interplay between bile acids and the gut microbiota itself. When thrown out of equilibrium, both higher and lower than normal bile acid levels can lead to undesirable health effects. The present analysis further demonstrates the close relationship between bile acids and microbiota composition, and the associations between fecal bile acids and individual microbial subgroups may give insight into future research regarding which taxa are of importance in bile acid metabolism and how groups of bacteria are modulated by fecal bile acids. In particular, we identified a trend where higher levels of bile acids in feces were positively associated with microbial subgroups previously identified as being associated with the risk of metabolic diseases or risk factors, and were inversely associated with microbial subgroups previously identified as being positively associated with higher diet quality and inversely associated with metabolic diseases or risk factors. Our findings indicate that further investigation is urgently needed regarding exactly how bile acid levels affect microbiota composition (e.g., which taxa, species) and how this interplay affects metabolic diseases and risk factors.

Fecal cholesterol and coprostanol levels were differentially associated with the microbial subgroups. As coprostanol is a microbial metabolite of fecal cholesterol, this is to be expected [38]. Fecal cholesterol levels were significantly positively associated with subgroups 6 and 13, subgroups which were previously associated with high serum triglycerides and BMI, respectively, and inversely associated with coprostanol [16]. This suggests that subgroups 6 and 13 do not have a high capacity for the metabolism of cholesterol to coprostanol. Conversely, we found that higher fecal coprostanol levels were positively associated with subgroups 1, 14, 16, 17, 18, and particularly 20. This indicates that these subgroups may be involved in cholesterol-to-coprostanol metabolism. As the production of coprostanol from cholesterol reduces the amount of cholesterol available for absorption from the gut, it may therefore lower serum cholesterol levels, and some research supports this notion [15]. Although no subgroup was previously significantly associated with total cholesterol or LDL-c levels, subgroups 1, 14, 16, 18, and 20 were previously inversely associated with metabolic diseases or risk factors such as waist circumference, serum triglycerides, serum HDL-c, and prevalent type 2 diabetes mellitus [16]. The positive associations identified here between fecal coprostanol and the subgroups suggest that cholesterol-to-coprostanol metabolism could potentially be an indirect mechanism contributing to the previously identified favorable associations with metabolic diseases or risk factors and these subgroups. 

Interestingly, the two subgroups (6 and 13) which were positively associated with fecal cholesterol were also positively associated with several plant sterols, namely stigmasterol, campesterol, and beta-sitosterol. Plant sterols are known to increase the excretion of cholesterol, while plant sterols themselves are poorly absorbed in humans [14]. This effect may explain why subgroups associated with fecal cholesterol are also associated with the main fecal plant sterols. However, our results indicate a differential association between sitostanol, a hydrogenated plant sterol, and the other plant sterols and the subgroups. This indicates that sitostanol may interact differently with the gut microbiota. Unfortunately, the microbial transformation of plant sterols has not been well studied and little information exists regarding the effect of plant sterol consumption or supplementation on microbiota composition [14,39]. Preliminary research indicates that plant sterols may affect microbial transformation of cholesterol or production of short-chain fatty acids, and may also impact microbiota composition [40]. Our results indicate that plant sterols and microbiota composition are an interesting avenue for future research. Strengths of this study include the large, well-characterized study population and the integration of both microbiome and metabolomics data. The use of Metabolon’s high-quality metabolite analysis process and identification platform is a further strength. Limitations of this study include that the analysis is cross-sectional and that only one stool sample per individual was available. Additionally, as causality cannot be determined by the design of our study, directionality of the associations cannot be interpreted. 

## 5. Conclusions

In this analysis, we investigated the relationship between fecal bile acids and sterols and twenty microbial subgroups previously identified in the KORA FF4 study. We identified a number of microbial subgroups which were consistently either positively or negatively associated with fecal primary and/or secondary bile acids. The two of the three subgroups which were positively associated with bile acids were also significantly positively associated with fecal cholesterol. Most of the subgroups that were associated with bile acids in the present study were identified in a previous analysis of ours as being associated with habitual diet and/or metabolic diseases or risk factors. These results indicate that future studies should seek to better understand the relationship between bile acids and the gut microbiota. Furthermore, our findings suggest that the microbial subgroups described here warrant further investigation in future studies, and also confirm LDA as a useful method to describe microbiota composition. 

## Figures and Tables

**Figure 1 metabolites-12-00846-f001:**
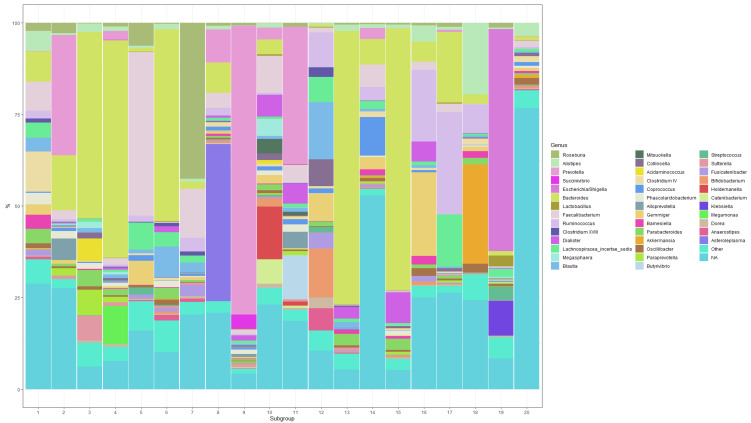
Composition of the 20 microbial subgroups at the genus level (showing the top 40 genera with a probability of at least 2% for at least one subgroup).

**Figure 2 metabolites-12-00846-f002:**
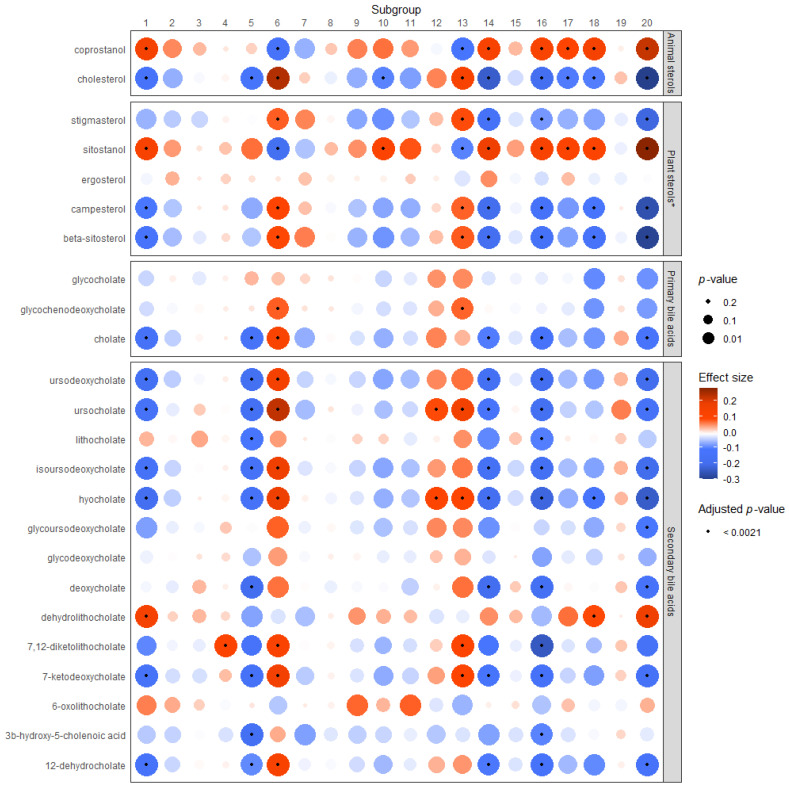
Associations between microbial subgroups and fecal bile acids or sterols. * The linear trend is shown here. However, these models indicate the presence of a non-linear relationship, which cannot be represented in the heat map. Appendix A depict the non-linear relationship for each plant sterol and subgroup in comparison to the linear trend shown here.

**Table 1 metabolites-12-00846-t001:** Characteristics of the study population by sex.

	Total		Men		Women	
	*n* = 1280		*n* = 636		*n* = 634	
*Continuous variables*	*Mean*	*SD*	*Mean*	*SD*	*Mean*	*SD*
Age (years)	59.72	12.03	59.87	12.30	59.57	11.76
Waist circumference (cm)	96.91	13.80	102.21	11.45	91.51	13.90
BMI (kg/m^2^)	27.8	4.8	28.0	4.0	27.5	5.5
Total energy (kcal/d) ^1^	1872.95	403.23	2120.43	351.85	1621.17	276.48
Total fiber (g/d) ^1^	17.89	5.07	18.39	5.15	17.39	4.94
Alcohol (g/d) ^1^	10.31	10.34	16.21	11.10	4.31	4.35
*Categorical variables*	*%*	*n*	*%*	*n*	*%*	*n*
Education:						
<13 years	63.9	818	58.2	376	69.7	442
≥13 years	36.1	462	41.8	270	30.3	192
Physical activity:						
Active	58	742	55.7	360	60.3	382
Inactive	42	538	44.3	286	39.7	252
Smoking:						
Current	15.4	197	16.6	107	14.2	90
Ex	35.9	459	42.1	272	29.5	187
Never	48.8	624	41.3	267	56.3	357

^1^ *n* = 930 total, *n* = 469 for men, *n* = 461 for women.

**Table 2 metabolites-12-00846-t002:** Metabolite Summary Statistics.

Metabolite	Min	25th %ile	Median	75th %ile	Max	% Missing
stigmasterol	0.002	0.033	0.051	0.078	0.663	4.53
sitostanol	0.001	0.029	0.052	0.079	0.441	7.42
beta-sitosterol	0.001	0.030	0.051	0.101	0.932	0.31
campesterol	0.001	0.028	0.051	0.104	1.139	1.17
ergosterol	0.001	0.025	0.054	0.119	10.336	4.21
cholesterol	0.002	0.024	0.055	0.142	1.507	0.23
coprostanol	0.0005	0.030	0.054	0.086	0.530	4.53
cholate	0.0002	0.015	0.051	0.210	64.810	3.20
glycochenodeoxycholate	0.003	0.024	0.052	0.129	12.007	6.02
glycocholate	0.001	0.021	0.052	0.152	12.721	1.09
12-dehydrocholate	0.002	0.017	0.053	0.225	61.283	17.42
3b-hydroxy-5-cholenoic acid	0.001	0.031	0.053	0.090	1.066	12.58
6-oxolithocholate	0.002	0.030	0.052	0.089	0.738	14.45
7,12-diketolithocholate	0.001	0.023	0.051	0.129	23.395	21.25
7-ketodeoxycholate	0.001	0.018	0.053	0.207	53.246	10.31
dehydrolithocholate	0.0004	0.024	0.051	0.092	0.465	1.41
deoxycholate	0.0003	0.020	0.054	0.110	1.809	1.95
glycodeoxycholate	0.001	0.022	0.052	0.129	11.315	8.83
glycoursodeoxycholate	0.003	0.025	0.055	0.135	6.540	18.36
hyocholate	0.002	0.028	0.052	0.106	2.646	9.22
isoursodeoxycholate	0.001	0.023	0.052	0.135	8.105	0.47
lithocholate	0.001	0.031	0.051	0.082	0.723	1.09
ursocholate	0.002	0.025	0.050	0.195	54.447	0.63
ursodeoxycholate	0.001	0.021	0.048	0.125	6.654	1.64

## Data Availability

The dataset analyzed in this study is not publicly available due to restrictions imposed by the Ethics Committee of the Bavarian Medical Association to protect the privacy of the study participants. However, a request for use of the data can be made via a project agreement through the KORA.PASST platform (https://helmholtz-muenchen.managed-otrs.com/external/).

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
