# Peer review of "Fecal Bile Acids and Neutral Sterols Are Associated with Latent Microbial Subgroups in the Human Gut"

_metabolites, 2022, doi:10.3390/metabo12090846_

Round 1

Reviewer 1 Report

The paper is interesting and continues previous research by the group of authors. My nly concern is the lack of mechanisms, in the discussion section, relating the association between some microbiota subgroups and healthier metabolic outcome. In the absence of this, I find that solely presenting associations is of limited value, nowadays

Author Response

Thank you very much for your time and careful review of our manuscript. Please see our response to your comments in the attachment.

Reviewer 2 Report

This an interesting study with adequate sample size for this analysis. Authors further need to address following comments:

1. Table-1: Any reported gastrointestinal complication in the study subjects?

2. Did you examine a correlation between biliary calprotectin and microbial subgroups population?

Author Response

(The authors gave the same response as above.)

Reviewer 3 Report

The authors created 20 subgroups based on the composition of the intestinal microbiota in the feces of the subjects, and the composition of the individual's intestinal microbiota was presented as a percentage of the subgroup combination. The authors then examined the correlation between the subgroups that showed the composition of the intestinal microflora and the metabolites bile acids and lipids in the feces of the subjects. This paper is an interesting and novel approach to the composition of the intestinal microbiota and disease. However, I have several concerns that need to be addressed before considering publication.

1.    In the present study, the author specified the composition of the intestinal microbiota as a combination of pre-determined subgroups, and the intestinal microbiota of individual subjects will be expressed as a percentage of multiple subgroups. In this case, is it possible that there are more than two combinations of subgroups that represent the intestinal microbiota of one person?

2.    The subjects in this study were also interviewed about their drug history. Was the subject also interviewed for medications such as ursodeoxycholic acid and elobixibat, which are involved in bile acid metabolism, and medications involved in lipid metabolism? Were subjects who were taking such drugs excluded from the study? If the authors did not exclude them from the subject population, what impact on the study results could be considered?

3.    In a previous study, the authors examined the subject's intestinal microbiota and lipid metabolism composition in the blood. What is the relationship between lipid metabolites in feces and blood in the present study? Did subjects with high fecal lipids have similar combinations of gut microbiota subgroups to subjects with blood lipid abnormalities?

4.    Are the 20 subgroups of gut microbiota selected for this study equally available for different populations under consideration? Or is it necessary to create new subgroups when the population differs?

Author Response

(The authors gave the same response as above.)

Reviewer 4 Report

The paper “Fecal bile acids are associated with latent microbial subgroups in the human gut” of Breuninger et al. investigates the relationship between fecal sterols and gut microbiome.  Fecal samples of over 1000 participants were analyzed by means of UPLC-MS/MS metabolomics, while the microbial composition was assessed by 16S rRNA sequencing. Using previously published approach individual “microbial fingerprints” were constructed bysed on the presence of 20 microbial subgroups. The topic of this study is actual and scientifically important, methodology appropriate (although I do not feel qualified to assess the microbial part of the methodology) and clearly described. The large number of study participants makes this study unique. Results are interesting and will definitely be of interest of readers/researchers in the field. However, due to the experimental setup, causality  of individual observations can not be determined and the study needs to be considered as explorative and descriptive (authors are aware of this fact and mention this as study limitation).

As a reader, I would welcome a more detailed discussion, even if this is (for abovementioned reasons) rather speculation. Eg-the authors show, that “good” microbial subgroups 5, 14, and 16 correlate with  decrease in both primary and secondary BA. Why it can be – these groups actually seem to prefer low sterols (except from ergosterol, virtually all sterols are lower). Where all the sterols disappear? Is it the function of the diet, host metabolism or bacterial effect (as degradation of sterols is quite unlikely in the human gut, the latter is not that probable).

I have just few minor comments:

·         Did study subjects use medication, that may affect bile acid/cholesterol metabolism? It is not clear, whether their use was exclusion criterion (ursodeoxycholic acid, obeticholic acid, bile acid sequestrants, statins, fibrates, pro/prebiotics,…).

·         Aimed the metabolomic study solely on steroids or was is a true untargeted metabolome? What were the parameters for data collection and analysis (m/z range,…)

·         Did you try to analyze serum/plasma samples to see, whether the fecal signature is somehow reflected in systemic circulation?

·         Systematic name of each compound may help non-bile acid experts to orientate in the structures (would be ok to include those eg in supplementary material). Also, stick to a single nomenclature and do not mix “chemical synonyms” – some compounds are named using “keto” prefix, while for other compounds either “oxo” or “dehydro” term is used instead.

Author Response

(The authors gave the same response as above.)

Round 2

Reviewer 3 Report

The authors have responded well to the reviewer's comments. I have no further comments.